# Evaluation of Antibacterial Properties of Zinc Oxide Nanoparticles Against Bacteria Isolated from Animal Wounds

**DOI:** 10.3390/pharmaceutics17020209

**Published:** 2025-02-06

**Authors:** Noppason Pangprasit, Aphisek Kongkaew, Duanghatai Saipinta, Surachai Pikulkaew, Montira Intanon, Witaya Suriyasathaporn, Wasana Chaisri

**Affiliations:** 1Akkhraratchakumari Veterinary College, Walailak University, Nakhon Si Thammarat 80160, Thailand; p.noppason@gmail.com; 2PhD’s Degree Program, Faculty of Veterinary Medicine, Chiang Mai University, Chiang Mai 50100, Thailand; 3Research Administration Section, Faculty of Medicine, Chiang Mai University, Chiang Mai 50200, Thailand; aphisek.k@cmu.ac.th; 4School of Veterinary Medicine, Faculty of Veterinary Medicine, Chiang Mai University, Chiang Mai 50100, Thailand; duanghathai.s@cmu.ac.th (D.S.); surachai.pikul@cmu.ac.th (S.P.); montiraintanon@gmail.com (M.I.); or suriyasathaporn.witaya.y3@f.mail.nagoya-u.ac.jp (W.S.); 5Research Center of Producing and Development of Products and Innovations for Animal Health and Production, Chiang Mai University, Chiang Mai 50100, Thailand; 6Cambodia Campus, Asian Satellite Campuses Institute, Nagoya University, Nagoya 464-8601, Japan

**Keywords:** antibiofilm, antimicrobial activities, wound, metallic nanoparticles, ZnO, Ag

## Abstract

**Background/Objectives:** This research aimed to determine the efficacy of metallic oxide nanoparticles, especially zinc oxide nanoparticles (ZnO-NPs), in inhibiting a wide range of bacteria isolated from animal wounds, indicating their potential as alternative antimicrobial therapies in veterinary medicine. **Method:** The disc diffusion technique, broth microdilution technique, and time-kill kinetic assay were performed to determine the antibacterial activity of the ZnO-NPs. **Results:** Transmission electron microscopy (TEM) and scanning electron microscopy (SEM showed that the ZnO-NPs were spherical and polygonal with sizes ranging from 50 to 100 nm, while DLS (NanoSizer) measured an average size of 512.3 to 535.7 nm with a polydispersity index (PDI) of 0.50 to 0.63 due to particle size agglomeration. The ZnO-NPs exhibited antibacterial activity against several bacterial strains isolated from animal wounds, including *Staphylococcus aureus*, *Staphylococcus epidermidis*, *Escherichia coli*, *Pseudomonas aeruginosa*, and *Klebsiella pneumoniae*, with inhibition zones ranging from 10.0 to 24.5 mm, average MIC values ranging from 1.87 ± 0.36 to 3.12 ± 0.62 mg/mL, and an optimum inhibitory effect against *Staphylococcus* spp. The time-kill kinetic assay revealed that the Zn-ONPs eradicated *Staphylococcus* spp. and *Klebsiella pneumoniae*, as well as *Escherichia coli* and *Pseudomonas aeruginosa* (99.9% or 3-log10 reduction), within 30 min of treatment. They also demonstrated a varying degree of antibiofilm formation activity, as indicated by the percentage reduction in biofilm formation compared to the untreated biofilm-forming bacterial strains. **Conclusion:** ZnO-NPs effectively inhibit bacterial growth and biofilm formation in animal wound isolates.

## 1. Introduction

Metallic and metallic oxide nanoparticles (MONPs) have recently emerged as promising antibacterial agents as an alternative to antibiotics for both Gram-positive and negative bacteria [1,2,3]. The antimicrobial efficacy of MONPs is intricately linked to their structural and physical properties, including size, shape, and zeta potential [4]. MONPs inhibit bacterial growth, release toxic metal ions, and produce reactive oxygen species (ROS); H_2_O_2_ and OH-, which can damage cell membranes and intracellular components, including nucleic acids and proteins [5]. Metallic nanoparticles such as aluminum, calcium, magnesium, silver, zinc, iron, copper, and titanium have been widely reported [6]. Zinc oxide (ZnO) is considered a potential antibacterial metallic oxide nanoparticle in biomedical applications due to its low cost, excellent biocompatibility, and long-term effectiveness, as well as the ability to change the morphology of ZnO nanoparticles using different synthesis routes, precursors, or materials [7]. Furthermore, compared to other metallic nanoparticles such as CuO, Ag and ZnO are less toxic to humans [8]. Previous studies have shown that ZnO-NPs have antibacterial action and biofilm inhibitory properties against a variety of pathogens, including *Streptococcus mutan*, *Clostridium* spp. [9], *Bacillus subtilis*, *S. aureus*, and *E. coli* [10]. In food and agriculture, ZnO-NPs have been used as artificial food additives and for the control of various microbial diseases in plants [11]. Regarding their therapeutic abilities, ZnO-NPs have attracted attention for application in the veterinary sciences. Nevertheless, knowledge about their application in veterinary medicine is limited.

Wounds are a common problem in small animal practice. The skin, due to its exposure to injuries, becomes more susceptible to being colonized by bacteria, which might potentially result in an infected wound [12]. These infected wounds often transform into chronic wounds characterized by delayed healing, thus increasing the cost of treatment [13]. Broad-spectrum antibiotics and antiseptics are typically employed to manage bacterial colonization at wound sites [14,15]. The ongoing administration of these treatments is contributing to an increase in resistance to antimicrobial agents in both the pathogenic bacteria and the commensal skin microflora [16]. This situation poses a risk for pet owners, as they may acquire antimicrobial-resistant bacteria and experience the transfer of antibiotic resistance genes to their own microbiota, which is emerging as a significant public health concern [17,18]. As a result, alternative antimicrobial agents, including metallic and metallic oxide nanoparticles, Ag-NPs, and TiO-NPs, are being investigated for use in infected wounds in animals [3]. ZnO-NPs are naturally known for their outstanding ability to inhibit and eliminate resistant bacteria [19]. Studying ZnO-NPs’ antibacterial activity against bacteria colonizing animal wounds is critical for developing effective techniques to control bacterial infections at wound sites. The objective of the present study was to examine the antimicrobial and antibiofilm-forming properties of ZnO-NPs against various bacterial strains obtained from animal wound sites.

## 2. Materials and Methods

### 2.1. Ethical Approval

This study received approval from the Animal Care and Use Committee, Faculty of Veterinary Medicine, Chiang Mai University (FVM-ACUC: S21/2565).

### 2.2. Study Period and Area

This experimental study was conducted between October 2022 and April 2023 and included total wound infection cases in companion animals who received wound therapy at a small animal clinic in northern Thailand and a small animal hospital/clinic in southern Thailand. All cases with unspecified wound types were included and swabbed using the Stuart Transport Medium (Thermo Scientific^TM^, Waltham, MA, USA). The swabbing technique was followed by Levine-like swab techniques [20]. The samples were stored at 4 °C and transported to the Veterinary Diagnostic Laboratory at Chiang Mai University, where they were cultured within 24 h.

### 2.3. Isolation and Identification of the Wound Bacteria

The swab samples were plated on selective and non-selective media, including a MacConkey agar for the Gram-negative bacteria, a Mannitol salt agar for *Staphylococci*, an Edward medium agar for *Streptococci*, and blood agar for fastidious organisms, and incubated aerobically at 37 °C for 24–48 h. The preliminary identification of bacteria was based on the colony characteristics. Bacterial contamination, defined as the presence of more than three distinct morphologies on non-selective media, was excluded from this study. All bacterial isolates were confirmed at the species level using a MALDI-TOF mass spectrometer (Bruker Daltonik, Coventry, UK) by identifying bacterial fingerprints based on intact ribosomal proteins according to the manufacturer’s recommendations. Spectra were analyzed using Bruker Biotype software and Real-Time Classification 3.0 software, as described by Schumann [21]. The isolates possessed a species-level identification score of ≤2.00, indicating a highly possible identification according to the manufacturer’s criteria.

### 2.4. Characterization of Metallic Nanoparticles (ZnO-NPs and Ag-NPs)

The ZnO-NPs and Ag-NPs were obtained from Sigma-Aldrich^®^, with the production number 793361 (Saint Louis, MO 63103, USA). TEM (JEOL JEM-2100, Tokyo, Japan) and SEM (FEI Apreo, Eindhoven, The Netherlands) were used to observe the morphology of the ZnO-NPs. The stock solution of the ZnO-NPs was dispersed using distilled water and sonicated for 5–10 min before being analyzed. Dynamic light scattering (DLS) using a NanoSizer (HORIBA SZ-100-S, Tokyo, Japan) was employed to determine the particle size and size distribution of the ZnO-NP solution. Certain metallic oxide compositions were measured using X-ray fluorescence spectrometry with the HORIBA XGT 5200WR (HORIBA, Tokyo, Japan). The particle size distribution was measured using laser diffraction with the HELOS (H4542) and QUIXEL software and PAQXOS 5.2 (Sympatec GmbH, 38678 Clausthal-Zellerfeld, Germany).

### 2.5. Antimicrobial Activity

#### 2.5.1. Agar Well Diffusion Methods

The agar well diffusion method described by Balouiri [22] was used to determine the antibacterial activity of MONPs. Five strains of *S. aureus*, *S. epidermidis*, *E. coli*, *P. aeruginosa*, and *K. pneumoniae*, along with reference strains of these bacteria, including *S. aureus* ATCC25923, *S. epidermidis* ATCC12228, *E. coli* ATCC25922, and *P. aeruginosa* ATCC27853, were used. The colony of each tested bacteria was suspended in a 0.85% normal saline solution and adjusted to 10^7^–10^8^ CFU/mL (McFarland density = 0.5). The bacterial suspensions were then spread onto Mueller–Hinton agar (MHA, Himedia^TM^, Thane, India) with a sterile cotton swab. Wells were subsequently created using a sterile cork-borer (6 mm in diameter). After that, 40 µL of the ZnO-NPs solution and Ag-NPs as a control at the same concentration (20.0 mg/mL) were added and incubated at 37 °C for 18 to 24 h. The antimicrobial activity of both metallic nanoparticle solutions was determined by measuring the zone of inhibition with a vernier caliper. The study was performed in triplicate.

#### 2.5.2. Broth Microdilution Method

The broth microdilution method described by Clinical Laboratory and Standards Institute (CLSI) guidelines [23] was used to determine the MIC and MBC values of ZnO-NPs and Ag-NPs. The ZnO-NPs and Ag-NPs were suspended in distilled water, and two serial dilutions were performed with tryptic soy broth (TSB, Himedia^TM^, India) to achieve concentrations ranging from 20.0 to 0.078 mg/mL. To achieve a final inoculum concentration of 10^5^ CFU/mL, 10 µL of the bacterial suspension (McFarland density = 0.5) was added to each well of a 96-well plate and incubated at 37 °C for 18–24 h. The minimal inhibitory concentrations (MICs) of ZnO-NPs were defined as the lowest concentration that inhibited visible bacterial growth in the broth. For minimal bactericidal concentrations (MBCs), 10 µL of the transparent medium was smeared on the plate count agar (PCA, Oxoid, Thermo Scientific^TM^, England) using the drop plate technique and incubated at 37 °C for 24 h. The MBC was defined as the lowest concentration of the ZnO-NP solution that inhibits the growth of bacteria on the agar medium. The tests were performed and duplicated (Figure 1).

#### 2.5.3. Time-Kill Kinetic Assay

Time-dependent killing capacities of the ZnO-NPs were determined by the method described by Appiah [24] with some modification. In brief, the bacterial suspension was adjusted to a concentration of 10^5^–10^6^ CFU/mL in Muller–Hinton broth (MHB, Oxoid, Thermo Scientific^TM^, England) containing ZnO-NPs at a concentration of 10.0 mg/mL based on previous antimicrobial activities, and time exposed for 0, 30, and 60 min and 4, 8, 12, and 24 h at 37 °C. Following each exposure period, the samples were serially diluted 10-fold in phosphate-buffered saline (PBS, Himedia^TM^, India), and 100 µL of each dilution was aseptically inoculated onto the standard plate count agar (PCA, Oxoid, Thermo Scientific^TM^, England) using the drop plate technique. The plate with the lowest dilution showing a countable number of bacterial colonies (25–250 CFU per inoculation) was selected for enumeration. The average CFU count from the triplicate plates was calculated (Figure 2). 

### 2.6. Antibiofilm-Forming Activity

The antibiofilm-forming activity was measured using the crystal violet assay described by Stepanović with some modifications [25]. In each well of a flat-bottom polystyrene 96-well microtiter plate, 90 µL of tryptic soy broth (TSB; Himedia^TM^, India) containing 1.0% glucose was added as a negative control. The same volume of 10 mg/mL ZnO-NP solution and Ag-NPs in TSB served as the treatment and positive control, respectively. All wells were inoculated with 10 µL of the bacterial suspension, including *S. aureus*, *S. epidermidis*, *E. coli*, *K. pneumoniae*, and *P. aeruginosa* (0.5 McFarland scale: 10^7^–10^8^ CFU/mL) and incubated at 37 °C for 24 h. Following incubation, each well was discarded and rinsed three times with phosphate-buffered saline (PBS, pH 7.2) to eliminate non-adherent cells. The plates were then air dried for 30–45 min. Subsequently, the adherent cell was fixed with 100 µL of absolute methyl alcohol for 20 min. Afterward, 200 µL of 1.0% crystal violet was added and incubated for 15 min in the dark. After removing the crystal violet dye and rinsing with sterile distilled water, the retained dye was solubilized with 0.5% (*v*/*v*) ethanol. The optical density (OD) of the adherent biofilm was measured using the absorbance of crystal violet at a wavelength of 570 nm in a microtiter plate reader (TECAN Sunrise^TM^, Männedorf, Switzerland). The biofilm formation capability of each bacteria is shown in Table 1. The percentage of biofilm inhibition was determined using Formula (1).(1)$$Biofilm\;reduction\;\%=\frac{{\mathrm{OD}}_{\mathrm{bacteria}}-{\mathrm{OD}}_{\mathrm{treatment}}}{{\mathrm{OD}}_{\mathrm{bacteria}}}\;\times \;100\%$$

### 2.7. Statistical Analysis

Descriptive statistics, including the mean, standard deviation (SD), and range, were generated using SPSS statistic software. Comparisons of the antibacterial and antibiofilm-forming activities of the MONPs between the groups were conducted using the Mann–Whitney U-test when normality was not met and the independent *t*-test when normality was met. Differences were regarded as statistically significant at *p* < 0.05.

## 3. Results

### 3.1. Characterization of ZnO-NPs

Figure 3 displays the morphology of ZnO-NPs determined by transmission electron microscopy (TEM) and scanning electron microscopy (SEM). ZnO-NPs were found to be polygonal and spherical in shape, with particle sizes less than 200 nm. However, the average particle sizes of the ZnO-NPs determined by the NanoSizer^®^ (HORIBA, SZ-100-S, Tokyo, Japan) ranged from 512.3 to 535.7 nm, with a polydispersity index (PDI) ranging from 0.503 to 0.633, while the Ag-NPs exhibited average particle sizes ranging from 107.2 to 117.9 nm, with a PDI of 0.30–0.309. These results indicate that the ZnO-NPs exhibited lower particle distribution and higher agglomeration compared to the Ag-NPs. The composition of the ZnO-NPs was confirmed via X-ray fluorescence spectrometry. The results demonstrated high material purity, with particles consisting only of zinc (Zn) and oxide (O) groups (Figure 4).

### 3.2. Antimicrobial Activities

Figure 5 shows the inhibition zone of the ZnO-NPs compared to the Ag-NPs against bacteria isolated from animal wounds. The results show that the ZnO-NPs at 20.0 mg/mL had antibacterial properties against all tested microorganisms and were comparable when compared to that of the Ag-NPs (*p* > 0.05). The average clear zones of the ZnO-NPs against *E. coli*, *K. pneumoniae*, *P. aeruginosa*, *S. aureus*, and *S. epidermidis* were 15.3 ± 0.54, 15.5 ± 0.57, 13.7 ± 1.11, 18.1 ± 0.83, and 20.1 ± 1.40 mm, respectively.

The minimum inhibition concentration (MIC) and minimum bactericidal concentration (MBC) values of the ZnO-NPs and Ag-NPs against skin microbial pathogens are displayed in Table 2. The results demonstrate that both MONPs exhibited bacteriostatic effects against all bacterial strains isolated from wounds, with MIC values ranging from 1.9 ± 0.4 to 15.0 ± 2.9 mg/mL. The effectiveness of both MONPs against Gram-positive bacteria was better than against Gram-negative bacteria, with lower MIC values ranging from 1.9 ± 0.4 to 3.1 ± 0.6 mg/mL. The MICs and MBCs of the ZnO-NPs against *S. aureus*, *S. epidermidis*, and *K. pneumoniae* were comparable to those of the Ag-NPs. However, for gram-negative bacteria, particularly *E. coli* and *P. aeruginosa*, ZnO-NPs exhibited lower efficacy, with higher MICs and MBCs values exceeding 10.0 and 20.0 mg/mL, respectively.

The time-kill assays were used to evaluate bacteriostatic and bactericidal activities of the ZnO-NPs. Declines in bacterial numbers of 3 logs of colony-forming units per milliliter (CFU/mL) or greater from the initial inoculation concentration were considered indicative of bactericidal action. The results show that the ZnO-NP solution at a concentration of 10.0 mg/mL demonstrated a significant bactericidal effect against Gram-positive bacteria, including *S. aureus* and *S. epidermidis*, reducing the initial log CFU/mL by over 3 logs following exposure (99% reduction) (Figure 6). There was complete inhibition of growth after exposure from 0 to 30 min. For *E. coli* and *P. aeruginosa*, the ZnO-NPs exhibited a bacteriostatic effect by simply lowering bacterial numbers immediately after exposure; nonetheless, the bacteria continued to multiply after 30 min. In contrast, the Ag-NPs exhibited complete inhibition of bacterial growth after exposure from 0 to 30 min against all tested microorganisms at the same concentration of ZnO-NPs.

### 3.3. Antibiofilm-Forming Activity

The crystal violet assay results show that all examined bacteria were biofilm formers.

Among the field and reference skin microbial pathogens, *S. aureus*, *S. epidermidis*, *E. coli*, *K. pneumoniae*, and *P. aeruginosa* were classified as bacterial biofilm former strains, whereas *S. aureus* and *S. epidermidis* were classified as strong-producing biofilm formers, *P. aeruginosa* was classified as a moderate-producing biofilm former (2ODNC < OD < 4ODNC), and *E. coli* and *K. pneumoniae* were classified as weak-producing biofilm formers (ODNC < OD < 2ODNC).

After treatment with MONPs, the percentage of biofilm inhibition was evaluated. The results demonstrate that ZnO-NPs and Ag-NPs were efficient antibiofilm agents against all tested strains, affecting the formation of biofilm and minimizing early cell adhesion when compared to the negative control (*p* < 0.05) (Figure 7). The ZnO-NPs exhibited less effective antibiofilm activity against *S. aureus* and *S. epidermidis* compared to the Ag-NPs; however, no significant differences were detected between the ZnO-NPs and Ag-NPs for the other bacterial species.

## 4. Discussion

The ZnO-NPs exhibited antimicrobial activity against all reference and field strains obtained from animal wound sites, showing various diameters of growth inhibition. The findings demonstrated that higher concentrations of ZnO-NPs were more efficient in producing inhibition zones of a larger diameter, indicating that the antimicrobial activities of the ZnO-NPs are concentration-dependent [26], and their effectiveness can vary significantly among specific strains, similarly to that of Ag-NPs. Several factors contribute to ZnO-NPs’ antibacterial activity, but the mechanism is still unclear and controversial [27]. However, various works in the literature provide several possible mechanisms. ZnO-NPs are capable of damaging bacterial cell walls, releasing antimicrobial ions (primarily Zn+ ions), inhibiting glycolytic enzymes through thiol group oxidation, and producing reactive oxygen species (ROS), such as superoxide anions (O_2−_) and hydroxyl radicals, which can damage bacterial cell membranes and interfere with protein synthesis and DNA replication, causing bacterial cell death [28]. Antibacterial activity can vary according to the medium’s components and the physicochemical properties of ZnO-NPs, such as their morphology, size, shape, concentration, and duration of contact with the bacterial cells [28,29,30]. The shape of the ZnO-NPs used in this study was spherical or polygonal, with the sizes ranging from 50 to 100 nm, as observed through SEM and TEM imaging; however, due to their aggregation, the particle sizes determined using the NanoSizer were larger (512.3–535.7 nm) [6]. The shape-dependent activity can be explained by the number of active facets. Facets with numerous atoms have stronger antibacterial properties [31]. The polar facets of ZnO-NPs contribute to their antibacterial activity, as these surfaces have a higher number of oxygen vacancies. Oxygen vacancies are known to boost the production of reactive oxygen species (ROS), which makes ZnO-NPs more effective at photocatalysis [32]. Conversely, some studies have suggested that oxygen deficiency, characterized by oxygen vacancies as the surface defects, is a critical factor in ROS production [33]. EDS analysis revealed that the ZnO sample consisted of 87.5 wt% Zinc (Zn) and 12.5 wt% oxygen (O), further supporting the presence of oxygen vacancies. Nevertheless, the antibacterial efficacy of ZnO-NPs in inhibiting bacterial growth is reduced in certain bacterial strains, particularly Gram-negative bacteria. This study found that *P. aeruginosa* exhibited high tolerance to ZnO-NPs. This reduced efficacy may be attributed to the specific characteristics of the bacterial strain or its ability to mitigate oxidative stress, which could impair the antibacterial potential of ZnO-NPs.

Consistent with previous reports, the ZnO-NPs demonstrated wide efficacy in inhibiting the growth of the majority of bacterial strains, especially Gram-positive bacteria [34,35,36,37]. Consistent with the MIC and MBC results, the ZnO-NPs showed more pronounced activity against Gram-positive bacteria, including *S. aureus* and *S. epidermidis*, than against *E. coli* or *P. aeruginosa*. The observed differences can be attributed to the distinct cell wall structures of the bacterial groups. Gram-negative bacteria have an additional outer membrane containing lipopolysaccharide (LPS), which improves barrier properties and increases bacterial resistance to antibiotics [37,38]. In addition, cell physiology, metabolism, or degree of contact are also affected by antibacterial activities [35]. In contrast, the ZnO-NPs exhibited bactericidal properties against *K. pneumoniae*, which is likewise a Gram-negative bacterium. This can be explained by the bacterium acting as a weak-producing biofilm forming, rendering it more susceptible to ZnO-NPs.

The results of the present study demonstrated that the ZnO-NPs and Ag-NPs acted as antibiofilm agents in all of the tested strains. A previous study explained that the negative charge of carboxylic acid, residual phosphate of uronic acid, or metal-bound pyruvate, which are components of the biofilm matrix, interact with positively charged metal ions (Zn^2+^ or Ag^+^) via electrostatic forces, resulting in cell death [38,39,40]. Once the nanoparticles reach the biofilm boundary, they penetrate deep into the biofilm matrix and distribute through these pore spaces. In comparison to the ZnO-NPs, the Ag-NPs demonstrated more potent antibiofilm activity against the strong biofilm formers *S. aureus* and *S. epidermidis*. This can be explained by the smaller size of Ag-NPs, which are more prone to interact with the biofilm matrix due to their larger surface area [41].

## 5. Conclusions

In conclusion, our research indicates that ZnO-NPs demonstrate significant potential in inhibiting growth and biofilm formation among all bacteria isolated from animal wound sites, achieving inhibition rates of up to 95% at a concentration of 20 mg/mL. This highlights the efficacy of ZnO-NPs as an effective antimicrobial agent, offering a promising alternative to traditional antibiotics for reducing microbial loads and enhancing wound healing in animals. Our hypothesis that ZnO-NPs could offer a more targeted and efficient approach to wound management was validated through these findings. In comparison to the existing literature, this study showed a higher efficacy of ZnO-NPs, particularly in biofilm disruption, which has not been thoroughly explored in previous works. Future research on the long-term efficacy, safety, and possible clinical application of ZnO-NPs, especially in veterinary medicine, is required to validate their utilization as a potential treatment for infected animal wounds.

## Figures and Tables

**Figure 1 pharmaceutics-17-00209-f001:**
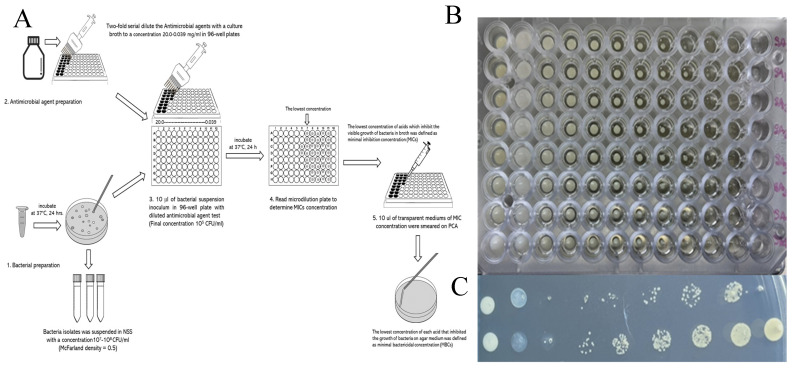
Broth microdilution method (**A**) and inhibition of bacterial growth in broth (**B**) and agar (**C**) by the ZnO-NPs against reference and field strains of bacteria.

**Figure 2 pharmaceutics-17-00209-f002:**
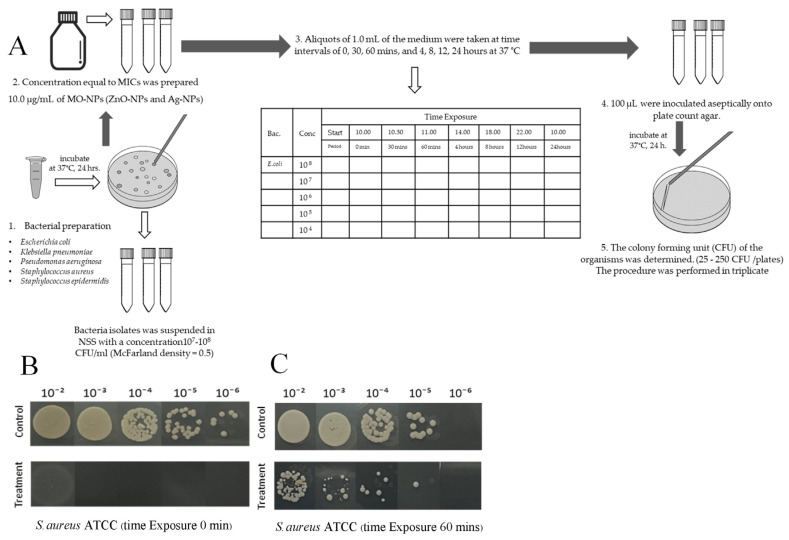
Time-kill kinetic assay of the ZnO-NPs against bacterial strains isolated from wound sites (**A**), including *S. aureus* ATCC25923, at different exposure times: 0 min (**B**) and 60 min (**C**).

**Figure 3 pharmaceutics-17-00209-f003:**
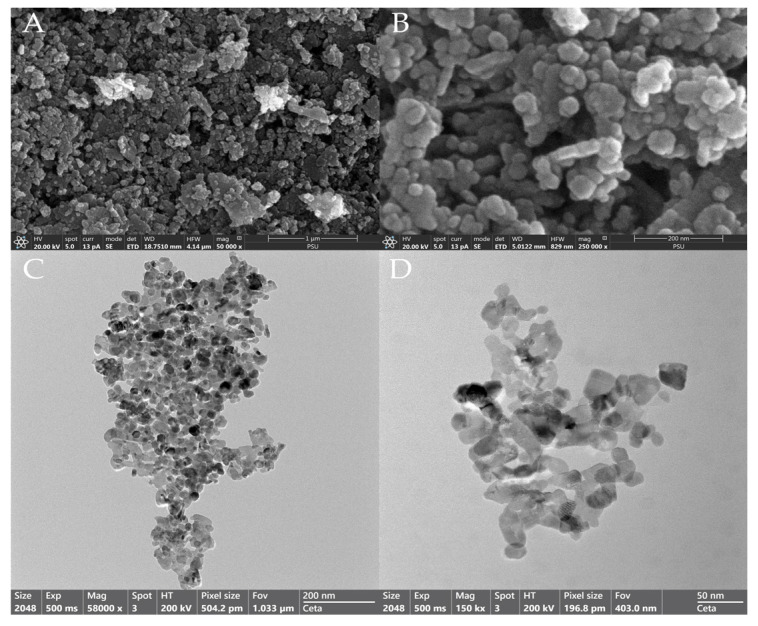
Scanning (**A**) bar 1 µm; (**B**) bar 200 nm and transmission electron microscopy (**C**) bar 200 nm; (**D**) bar 50 nm images of the ZnO-NPs.

**Figure 4 pharmaceutics-17-00209-f004:**
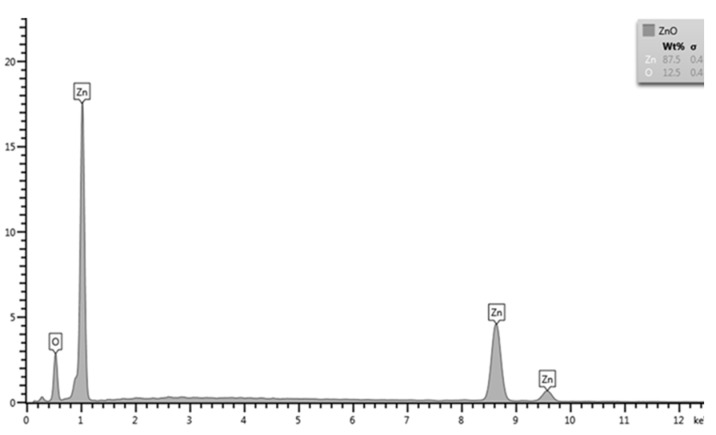
The composition of the ZnO-NPs, determined via X-ray fluorescence spectrometry.

**Figure 5 pharmaceutics-17-00209-f005:**
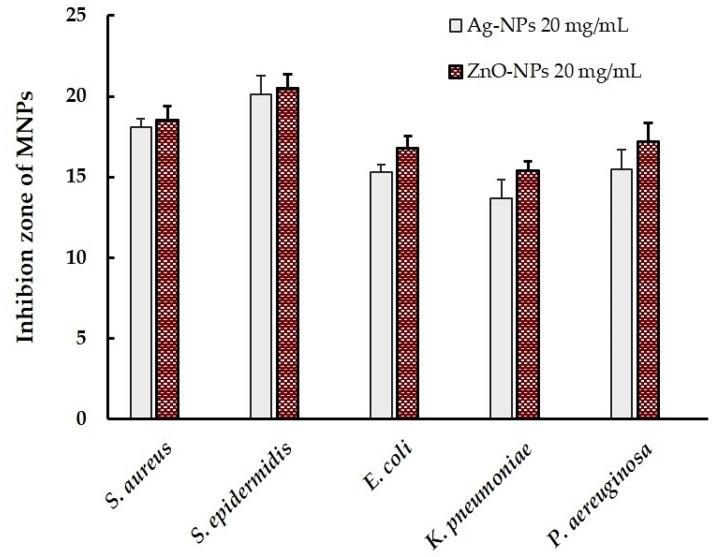
Average inhibition zone of the MNPs (ZnO-NPs and Ag-NPs) against bacteria isolated from animal wounds.

**Figure 6 pharmaceutics-17-00209-f006:**
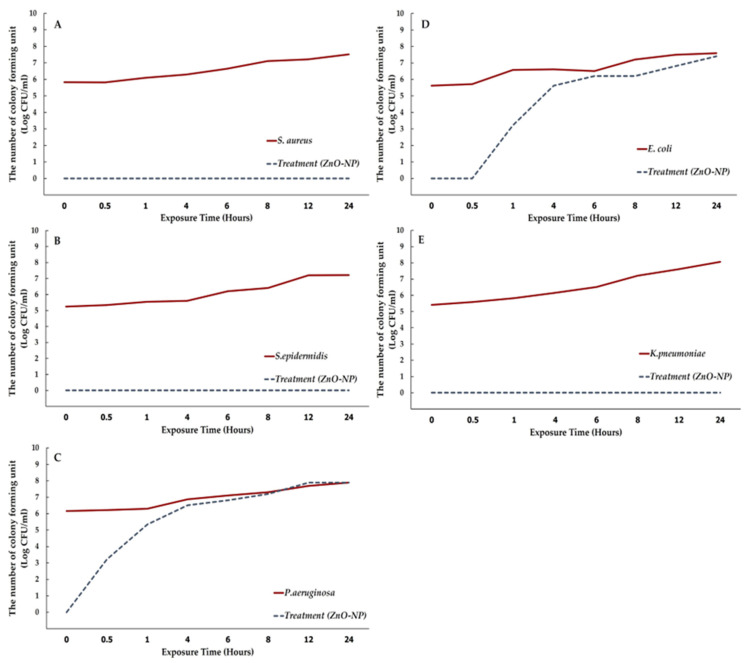
The time-kill kinetic profiles of the ZnO-NPs at a concentration of 10.0 mg/mL against bacterial strains, including *S. aureus* (**A**), *S. epidermidis* (**B**), *P. aeruginosa* (**C**), *E. coli* (**D**), and *K. pneumoniae* (**E**).

**Figure 7 pharmaceutics-17-00209-f007:**
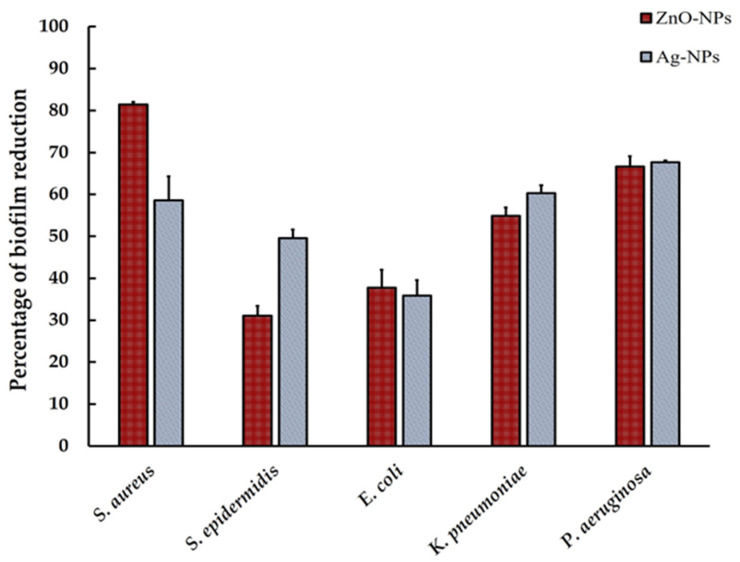
Percentage of biofilm reduction of the ZnO-NPs and Ag-NPs against skin microbial pathogens.

**Table 1 pharmaceutics-17-00209-t001:** Biofilm formation classification.

Cut-Off Value	Biofilm-Formation Classification
OD ≤ OD_NC_	Non-biofilm-formation
OD_NC_ < OD < 2OD_NC_	Weak-biofilm-formation
2OD_NC_ < OD < 4OD_NC_	Moderate-biofilm- formation
4OD_NC_ < OD	Strong-biofilm- formation

Abbreviations. OD_NC_: average OD of negative control +3 standard deviation of negative control; OD: OD of bacteria.

**Table 2 pharmaceutics-17-00209-t002:** Minimum inhibitory and bactericidal concentrations of the ZnO-NPs and Ag-NPs against skin microbial pathogens (five isolates of field and reference strains).

Bacterial Pathogens	ZnO-NPs (mg/mL)		Ag-NPs (mg/mL)	
	MICs	MBCs	MICs	MBCs
*E. coli* (Field strains)	10.0 ± 0.0	>20.0 ± 0.0	4.4 ± 0.6	7.5 ± 1.4
*E. coli* ATCC25922	10.0 ± 0.0	>20.0 ± 0.0	10.0 ± 0.0	10.0 ± 0.0
*K. pneumoniae* (Field strains)	3.8 ± 0.7	5.0 ± 0.0	3.8 ± 0.7	6.2± 1.3
*K. pneumoniae* (Reference)	5.0 ± 0.0	5.0 ± 0.0	5.0 ± 0.0	5.0 ± 0.0
*P. aeruginosa* (Field strains)	15.0 ± 2.9	>20.0 ± 0.0	3.8 ± 0.7	8.8 ± 1.3
*P. aeruginosa* ATCC27853	10.0 ± 0.0	>20.0 ± 0.0	10.0 ± 0.0	10.0 ± 0.0
*S. aureus* (Field strains)	1.9 ± 0.4	2.5 ± 0.0	2.5 ± 0.0	5.0 ± 0.0
*S. aureus* ATCC25923	2.5 ± 0.0	2.5 ± 0.0	2.5 ± 0.0	2.5 ± 0.0
*S. epidermidis* (Field strains)	2.5 ± 0.0	5.0 ± 0.0	3.1 ± 0.6	5.0 ± 0.0
*S. epidermidis* ATCC12228	2.5 ± 0.0	5.0 ± 0.0	2.5 ± 0.0	5.0 ± 0.0

## Data Availability

Data are contained within the article.

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
