# Peer review of "Evaluation of Antibacterial Properties of Zinc Oxide Nanoparticles Against Bacteria Isolated from Animal Wounds"

_pharmaceutics, 2025, doi:10.3390/pharmaceutics17020209_

Round 1

Reviewer 1 Report

Comments and Suggestions for Authors

The article "Evaluation of Antibacterial Properties of Zinc Oxide Nanoparticles Against Bacteria Isolated from Animal Wounds" describes the antibacterial activity of ZnO particles against some bacterial strains. While the article shows promise, it requires major revisions before it can be considered for publication in this journal:

Using DLS technique (NanoSizer) to determine the particle size in this case will give far larger values due to the nanoparticles agglomeration and because the DLS is reporting the hydrodynamic size of the agglomerate not the individual particle size. As the authors have SEM and TEM micrographs, they can report accurately the size of NPs from there. Therefore, the actual NPs size should be given in the abstract as this section is often presented separately in search engines and it must be able to stand alone as an informative piece.

Please use words rather than expressions as keywords – antibiofilm instead of antibiofilm forming ability, wound instead of animal wound, ZnO instead zinc oxide nanoparticles. For the last one as zinc oxide is in title is better to switch to ZnO in keywords to improve the hit chances. As authors have compared activities with silver NPs I suggest adding silver as keyword too.

Zinc oxide is a semiconductor, a metallic oxide compound. Therefore, the keyword “metallic nanoparticles” is misleading, as Ag, Au, Cu would correspond to it (is very important to respect the nomenclature as the literature is already signaling some worrisome trends in this domain). Metallic nanoparticles Ag, Au, Cu are one type of compound and oxide nanoparticles as ZnO, CuO, TiO2 are another type. This change is mandatory across the manuscript, from introduction to conclusions section.

The nanoparticle term is usually reserved for structures with at least one dimension smaller than 100 nm and if authors report size in excess of 500 nm then the structures should be named particles. At the same time a quick look on the TEM micrographs indicate that the particle size is under 50 nm (see figure 1D). Authors should remake section 2.1 with proper description of the SEM and TEM results (for example the particle size distribution can be made by measuring the individual particles with equipment software or free software as ImageJ. Additionally, in section 2.1 silver nanoparticles are introduced for comparison from nowhere – it is not clear why authors chose to compare ZnO with Ag NPs instead of standard antibiotics. Some background information on Ag NPs must be given in Introduction and Abstract sections to signal this kind of comparison. On the other hand, the antimicrobial activity of ZnO and Ag is well established and authors only tested commercially available samples (Sigma) against some bacterial strains, therefore the authors should indicate clearly which is the novelty of their study.

Please add some more studies about the antimicrobial activity of ZnO and Ag NPs. Previous work of Motelica et al on ZnO and Ag NPs can support the idea of this manuscript.

Figure 2 is not X-ray crystallography, but EDX (EDS) – Energy-dispersive X-ray spectroscopy and the mistake cast an important shadow on the scientific background. The composition % reported by the EDS indicate that the obtained ZnO sample is oxygen deficient, which means that it has oxygen vacancies. This is particularly important as the surface defects are the centers responsible for generation of ROS (reactive oxygen species) which are implicated in one of the antibacterial mechanisms of ZnO (see doi: 10.3390/pharmaceutics14122842 for further details on antibacterial possible mechanisms, including influence of the morphology –size and shape- on the antibacterial activity).

Please use proper indices and exponents (e.g. “O2-“ which should be O2- to match superoxide label, but which can be easily mistaken for O2- when is improper written).

The authors declare in section 4.4 that the ZnO NP were obtained from Sigma-Aldrich. Such commercial sample has some technical characteristics that should be given in a research article.

HORIBA XGT 5200WR – is a XRF device (X-ray fluorescence spectrometer) and cannot perform XRD analysis.

Abbreviations should be expanded upon first mention in the text (e.g. MIC, MBC, CFU etc), and only once not at every use which defies the need for an abbreviated form.

Use uniform notation for measurement units (now for litre are used both l and L like in figure 6 and figure 7, section 4.6. but also elsewhere across the manuscript). Personally, I would recommend the use of L.

Manuscripts published in prestigious journals must explain the significant advances provided in approaches and understanding compared to previous literature, and/or demonstrate convincingly potential in new applications. The Conclusions of your paper are especially important for this. Therefore, please try to sharpen this further. The optimal Conclusion should include:

• A summary of your key findings with actual numbers.

• A highlight of your hypothesis, new concepts, and innovations.

• A summary of key improvements compared to findings in the literature.

• Your vision for future work.

Author Response

Thank you very much for taking the time to review this manuscript Pharmaceutic-3413661. Please find the detailed responses below and the corresponding revisions/corrections highlighted (YELLOW) / in- track changes in the re-submitted files.

Reviewer 2 Report

Comments and Suggestions for Authors

The authors are advised to take these points into consideration:

Kindly specify the methods used to determine the zeta potential and polydispersity index (PDI) of the nanoparticles.

In the manuscript it is given that the MIC values for the metallic nanoparticles ranged from 1.9 ± 0.4 to 10.0 ± 0.0 mg/mL, however in the results maximum value is depicted as 15.0 ± 2.9 mg/mL against P. aeruginosa (Field strains). Please rectify this.

Lines no 75 “According to the advantage of ZnO-NPs, which is naturally known for its outstanding 75 microbe resistance.” Please correct this sentence

kindly use term either animal wounds or pet wounds for uniformity.

Comments on the Quality of English Language

Can be improved

Author Response

Thank you very much for taking the time to review this manuscript Pharmaceutic-3413661. Please find the detailed responses below and the corresponding revisions/corrections highlighted (BLUE) /in track changes in the re-submitted files.

Round 2

Reviewer 1 Report

Comments and Suggestions for Authors

The article “Evaluation of Antibacterial Properties of Zinc Oxide Nanoparticles Against Bacteria Isolated from Animal Wounds” seems well-revised and now can be recommended for publication.